# Slow Fashion Is Positively Linked to Consumers' Well-Being: Evidence from an Online Questionnaire Study in China

An Liu [1,2,*] , Emily Baines [2] and Lisbeth Ku [3,*]

1   School of Future Design, Beijing Normal University, Zhuhai 519087, China
2   School of Fashion and Textiles, Faculty of Arts, Design and Humanities, De Montfort University, Leicester LE1 9BH, UK
3   Division of Psychology, Faculty of Life Sciences, De Montfort University, Leicester LE1 9BH, UK
*   Correspondence: lis.ku@dmu.ac.uk (L.K.); liuan@bnu.edu.cn (A.L.)

**Abstract:** The environmental price of fashion has been heavily scrutinised in recent years. Slow fashion, with its emphasis on quality, design, sustainability, ethicality and local craft heritage, represents an alternative to the harmful environmental and social impact of fast fashion. Equally important, some initial evidence from qualitative research suggests that slow fashion could enhance consumers' well-being. The present study aims to quantitatively evaluate the relationships that fast and slow fashion may have with different domains of well-being, utilising Seligmen's influential PERMA model from positive psychology. In addition, it explores characteristics from slow fashion that may enhance garment lifetimes. An online questionnaire successfully surveyed 763 urban Chinese consumers. Results showed that consumption of slow fashion, in particular 'customised/bespoke clothing' that allows consumers to be actively involved in the creation process, positively predicted three well-being domains—engagement, meaning and achievement. Fast fashion, on the other hand, negatively predicted these domains. Classic/timeless design, ease of maintenance and ease of matching with other clothes emerged as the three most important characteristics that may encourage consumers' long-term use of fashion items. Implications of the findings are discussed in the context of promoting slow fashion to enhance sustainability.

**Keywords:** slow fashion; fast fashion; well-being; garment lifetimes; Chinese consumers

## 1. Introduction

The fashion industry has been under increasing scrutiny concerning its negative impact on the environment. It is the world's second most polluting industry, responsible for 10% of global $CO_2$ emissions and 20% of water waste [1,2]. The emergence of the fast fashion concept since the 1990s, a trend that relies on cheap production and frequent consumption [3], results in the production of 92 million tonnes of waste per year [4]. Clearly, changes in the fashion industry are urgently needed. These should not only focus on implementing sustainable practices in the supply chain, but should also facilitate fundamental shifts in consumption behaviour. Specifically, a switch back to 'slow' fashion—a practice that decreases clothing purchases and increases garment lifetimes [4]—is of the utmost importance.

In the past decade, research effort on sustainable practices in the supply chain management of the fashion industry has resulted in an invaluable volume of studies that examine diverse topics such as employing monitoring and auditing activities to ensure compliance of sustainability targets [5], green distribution and retailing [6] and the proposition of holistic roadmaps towards establishing sustainable supply chains [7]. On the other hand, how to change consumer behaviour such as diverting preference from fast fashion to slow fashion has received relatively little attention and needs further efforts. Indeed, a recent literature review by Domingos and colleagues [8] concludes that there are 'few published papers on Slow Fashion and even fewer on Slow Fashion consumer behaviour' (p. 12).

The present research project aims to fill this important gap by asking a general research question about how we could promote slow fashion among consumers in order to achieve long-term sustainability. Specifically, we propose two possible pathways. The first is through consumers' well-being, as an emerging body of evidence suggests that slow fashion may be conducive to higher levels of individual well-being [9,10]. Hence, the inclusion of well-being factors in the design and promotion processes of slow fashion might provide a pathway to influence consumer preference and ultimately to behavioural changes in consumption. The second pathway is through characteristics of slow fashion that are considered most important by consumers. As we discuss in the following, the concept of slow fashion encompasses many different characteristics. An examination of these characteristics from consumers' perspectives would help us understand which characteristics of slow fashion would be most conductive in prolonging garment lifetimes, thereby enhancing sustainability. Hence, the general research question is further divided into two specific research questions:

(1) How does slow fashion consumption relate to consumers' well-being?
(2) Which characteristics of slow fashion are most conductive in enhancing consumers' long-term use?

In the following, we first outline the theoretical frameworks for slow fashion (Section 2.1) and well-being (Section 2.2), and then review the literature on the possible relationships that slow and fast fashion may have with individual well-being (Section 3). In Section 4, we explain in detail the aims and objectives of the present research, along with a rationale for its hypotheses and methodology. After reporting our analysis strategies in Section 5, we report the results (Section 6), and discuss the implications (Section 7).

## 2. Theoretical Frameworks

### 2.1. Slow Fashion

The concept of slow fashion was first proposed by Fletcher [11], who asserts that slow fashion is about a shift from quantity to quality, and that it balances fashion expression with durability and long-term engagement. As slow fashion values local cultural heritage and craftsmanship, and incorporates social responsibility, sustainability and transparency to business practices [11], it has the potential to change the ways of producing, designing and consuming fashion [12]. By synthesising theorisations of slow fashion from different scholars, we propose slow fashion as a concept that encompasses the following five main factors:

#### 2.1.1. Quality

Slow fashion emphasises the importance of quality [11,13,14], focusing on finely crafted fashion items and the production cycle of clothing [13,15]. It reduces the negative environmental and social impact of over-consumption by combining high quality and durable fabrics with production processes that involve a high level of workmanship. It is therefore the opposite of the rapid, mass production approach of fast fashion.

#### 2.1.2. Design

Slow fashion aims to design clothing items that are aesthetically and stylistically durable and versatile [16]. so that consumers can wear them in a variety of ways to suit different needs. This is to aid slow consumption, a principal component of slow fashion [17]. Slow fashion design is less influenced by fashion trends, and uses timeless and versatile designs rather than trend-driven products. Therefore, classic designs, and functional designs which add to the utility of the product, are seen as the way to slow fashion.

#### 2.1.3. Sustainability

Most scholars [11–15] believe that slow fashion cares about environmental protection. It is a movement that inspires designers to use greener materials and more sustainable design methods, taking into account the laundering and maintenance of products, as well

as the environmental impact of garments at all stages of their life cycle, including when they are no longer being used or when they are being discarded [18]. Slow fashion is not only environmentally sustainable through the use of clean production techniques, but also by slowing down and reducing fashion production to reduce the raw materials used. This will ease the pressure on natural cycles and enable the regenerative capacity of the earth. Slow fashion takes an integrated approach that considers the interconnection of ecological and social systems in sustainable development [15,19,20].

### 2.1.4. Ethicality

Slow fashion is in line with ethical production and fair trade principles. Fast fashion's quest for speed, mass production and keeping retail prices competitive often means low wages and substandard working conditions for workers [21]. In contrast, slow fashion aims to ensure that workers are treated with the respect they deserve so that they can earn accordingly and benefit from the fashion value chain [16,17]. A slow production system adopts a human-centred perspective. It avoids excessive workloads and allows better working conditions for workers [16].

### 2.1.5. Local Cultural Heritage and Crafts

The slow fashion movement supports local communities and sustainable development, protecting diversity and cultures. It values the inheritance and application of traditional craft skills, and makes use of local expertise to enhance the value of fashion products. Scholars [12,16,17,22] argue that one of the development goals of slow fashion is to link skilled local craft production to international fashion markets to help workers overcome low wages and harsh working conditions, and to change the mass production model of fast fashion in order to benefit local craftspeople. Consequently, it can support sustainable development by introducing different regional cultures into the fashion industry and by promoting cultural and aesthetic diversity while maintaining economic benefits.

### 2.2. Well-Being

There are many different ways to define well-being. Apart from objective well-being, which can be indicated by factors such as physical and mental health, there is the concept of subjective well-being (SWB), which refers to individuals' own assessment of their happiness or well-being levels. Positive affect, or emotion, is commonly considered as one of the main indicators of SWB, but cognitive evaluation of one's life (such as whether one achieves one's aims, whether one sees meaning in one's life) is as important [23,24]. Indeed, from the perspective of positive psychology, well-being is not simply an absence of negative functions such as depression, loneliness or illness, but the presence of positive affect, social connection and wellness [25].

One of the oldest definitions of well-being is eudaimonic well-being, which can be traced back to Aristotle's concept of eudaimonia. According to Deci and Ryan [26], eudaimonic well-being 'is not so much an outcome or end state as it is a process of fulfilling or realising one's daimon or true nature—that is, of fulfilling one's virtuous potentials and living as one was inherently intended to live' (p. 2). Eudaimonic well-being is therefore often referred to as living the 'good' or 'purposeful' life [25]. This is in addition to another common definition of well-being—living a 'happy' life, which focuses on emotion and which is often referred to as hedonic well-being [26].

Incorporating both eudaimonic and hedonic components of well-being is the concept of 'flourishing', proposed by Seligman in his PERMA theory [25]. Seligman asserts that flourishing is the optimal state of psychosocial functioning, and, as such, represents the highest level of well-being and happiness. Instead of merely indicating a general level of well-being, flourishing is a multidimensional construct that has five different indicators: positive emotion (P), engagement (E), relationships (R), meaning (M) and accomplishment (A). As such, the PERMA model represents a synthesis of hedonic (P) and eudaimonic well-being (E, M, and A), along with a third dimension—relationships (R). A flourishing

individual exhibits frequent positive emotion, demonstrates a high level of psychological engagement with their work and their life, forms close and positive relationships with their family, friends and peers, feels that life is valuable, worthwhile and meaningful, and has a sense of working toward and reaching goals.

The five indicators of the PERMA model are interrelated, but nevertheless separate, as they assess well-being across multiple domains. The multidimensional approach does not only have theoretical relevance, but it also provides practical benefits as the separate indicators can shed light on meaningful variation amongst different domains. For example, if an individual scores particularly low on one indicator, intervention strategies can be developed to target improved functioning in that specific domain. As the present project seeks to explore the relationships that fast and slow fashion may have with well-being, the five indicators will be especially useful in shedding light on the possible psychological mechanisms that may underlie the general relationship (see [27] for a detailed discussion of the theoretical and practical benefits of a multidimensional approach to well-being).

### 3. Literature Review: Slow Fashion and Well-Being

Even though slow fashion is closely related to ecological well-being of the environment, which would inevitably affect individual well-being [28], currently there is a dearth of research on the issue. Indeed, of the 25 research papers on slow fashion that Domingos et al. [8] review, none of them examines the possible relationship between slow fashion and individual well-being. There is, however, some suggestion that fashion could potentially contribute to one's psychological well-being, as well as potentially having a negative impact. For example, Hefferon and Boniwell demonstrate that fashion tends to be associated with positive emotions [29]. Similarly, Hall [30] and Lotze [31] believe that the positive experience of fashion is manifested through the daily practice of styling, which can bring psychological comfort, satisfaction and happiness. This is in line with Horn's view that the positive emotions associated with clothing are conducive to a more positive self-perception and a higher level of self-worth [32]. Despite many possible positive relationships between fashion and individuals' well-being, fashion is also intricately linked to ill-being. Research on compulsive buying disorder, a type of behavioural addiction that is characterised by obsessive cognition about shopping and excessive buying behaviour that often leads to distress and/or impairment [33], show that compulsive buying behaviour often takes the form of individuals purchasing fashion items [34], and a strong fashion interest is an important predictor of the disorder [35,36]. Similarly, researchers working in the materialism paradigm show that fashion consciousness and fashion consumption are closely related to conspicuous consumption and status buying [37–39], and as a result could lead to overspending and debts [40].

Most studies do not explicitly discuss whether it is fast or slow fashion that is linked to these worrying relationships with materialism, compulsive buying and conspicuous consumption. However, the materialistic desire for ever more consumption coincides with fast fashion's business model of large selections and quick trend updates. A recent study [41] further shows that consumers' materialistic orientation directly affects purchase intention for unsustainable fashion products that are characteristic of fast fashion. Furthermore, individuals who exhibit compulsive buying tendencies often use image-related consumption such as buying clothes as a way to manage negative emotion and self-doubts [42]. All these suggest the relationship between fast fashion consumption and well-being would be a negative one.

On the other hand, elements of slow fashion are more readily and positively associated with well-being. For example, after consuming sustainable fashion, respondents reported a significant sense of personal growth, happiness and pleasure [8]. Some respondents reported feeling more comfortable and relaxed as they moved away from mainstream fashion and the competitive consumer pressure that came with it. As a result, they felt more confident about themselves [8].

The conscious act of reducing fashion consumption by focusing on quality, longevity and minimal design of fashion items has been proposed as a feasible way for consumers to remain fashionable while cutting down consumption [43]. To evaluate the impact of such reduction on consumers, Bardey et al. [44] study 10 women's lived experiences with a minimal wardrobe and report their participants experiencing reduced stress from fashion trends, with increased joy from fashion style.

A qualitative analysis by Masuch and Hefferon [10] shows that the attachment of nostalgia to clothing memorabilia is a way of constructing, reinforcing, negotiating and maintaining identity. It enables introspection through a process of positive recall and nostalgic recollection and allows one to pass through time and revisit meaningful social relationships. Buse and Twigg [45] also demonstrate the importance of old objects that hold memories. By examining the importance of handbags as objects of personal biography for people with Alzheimer's, the qualitative study shows that there is a sense of comfort and identity restoration in keeping old things when one's social role and identity is being challenged or changed.

In addition, customised/bespoke clothing gives consumers the opportunities to experience a sense of engagement and self-expression through their involvement in the creation processes [46]. Ryan and Deci's self-determination theory (SDT) proposes that humans are by nature active and growth-oriented [47]. The theory posits three basic psychological needs—autonomy (feeling volition in one's actions), competence (feeling capable) and relatedness (feeling connected to other people), and the resulting well-being when these needs are satisfied [47]. Customised/bespoke clothing allows for greater consumer autonomy than is normally the case in mass production. By enabling more personal preferences to be brought into the creation processes, this slow fashion type not only creates products that are better suited to consumers' desires, but it also provides opportunities for them to experience engagement. All this would contribute to the satisfaction of autonomy and competence needs. Therefore, it seems that slow fashion consumption is likely to have positive relationships with well-being as many factors that are integral to slow fashion, such as sense of identity, relationships, engagement and positive emotion, are all well documented predictors of well-being [48,49].

## 4. The Present Research

### 4.1. Aims, Rationale and Hypotheses

Despite the illuminating evidence reviewed above, a number of issues require consideration in the relationship between fast/slow fashion and well-being. The positive link between slow fashion and well-being needs to be evaluated in a more systematic way. The current literature is limited to mostly qualitative studies with small samples drawn from Western societies [10,50,51]. Not only does the relationship need to be evaluated with larger and more diverse samples, it also needs to be systematically examined in the context of different categories or types of slow fashion. As discussed above, slow fashion encompasses a number of different factors. While it is often considered that slow fashion focuses on quality [10,13–15], design [16,17], sustainability [11–20], ethicality [16,17,21] and local craft heritage [12,16,17,22], there is currently no one single definition. Many fashion companies in China, where the current research was conducted, may relate to some but not all of these specifications. Hence, the present study seeks to evaluate the relationships between different fashion types and well-being, taking into consideration that consumers' understanding of slow fashion may vary, as particular fashion types may embody certain elements of slow fashion. For instance, famous international luxury brands may represent the quality over quantity specification of slow fashion, but may or may not include the environmental sustainability, ethicality or local craft heritage elements. Likewise, eco clothing may fit the sustainability bill well, but does not necessarily see craft heritage as its main focus of effort. Due to the limited literature concerning the exact relationships, we are not in a position to form specific hypotheses concerning which type of slow fashion is related with which aspects of well-being. Hence, the current study adopts a general hypothesis that

slow fashion is positively related with well-being. Nevertheless, we acknowledge that this relationship might be different across different types of slow fashion and the methodology we adopted in our operationalisation of slow fashion, as explained below in Section 4.2, would allow us to explore these relationships in greater detail.

Second, like slow fashion, well-being is best considered as a multidimensional construct. As we delineate in Section 2.2, the five different domains of well-being as proposed by Seligmen are often interrelated, but they nevertheless represent distinctive components that may affect an individual's assessment of their well-being differently [24,25,52]. Hence, understanding well-being as a single construct may obscure important variations in the relationships between fashion and well-being. To use the two examples we discussed above—nostalgic clothing and customised/bespoke clothing—it is likely that the first is more closely related to the emotional, hedonic aspect of well-being, while the latter more with the cognitive, eudaimonic aspect of well-being. Hence, the present study adopts a multidimensional approach in our attempt to examine the relationships fast and slow fashion may have with well-being. Specifically, we differentiate the hedonic component of well-being from the eudaimonic ones and hypothesise:

**Hypothesis 1 (H1).** *Consumption of slow fashion is a positive predictor of hedonic well-being—'positive emotion'.*

**Hypothesis 2 (H2).** *Consumption of fast fashion is a negative predictor of hedonic well-being—'positive emotion'.*

**Hypothesis 3 (H3).** *Consumption of slow fashion is a positive predictor of eudaimonic well-being—'engagement', 'meaning' and 'achievement'.*

**Hypothesis 4 (H4).** *Consumption of fast fashion is a negative predictor of eudaimonic well-being—'engagement', 'meaning' and 'achievement'.*

We do not hypothesise about the third domain 'relationship' in Seligmen's PERMA model, as currently there is very little literature on this domain. Even though there are works related to the role of objects in relationships (e.g., Mauss's gift theory, see [53,54]), and artefacts as an intermediary in human relationships [55,56], these are not directly related to the social support sense of relationship proposed by Seligmen. Nevertheless, we retain the domain 'relationship' in our study as an exploratory part of our analyses.

Last but not least, the present study attempts to answer the second research question—Which characteristics of slow fashion are most conductive in enhancing consumers' long-term use?—by exploring consumers' evaluation of slow fashion characteristics. Based on the five main factors of slow fashion that we outlined in Section 2.1, 13 possible characteristics that may contribute to garment lifetimes were generated. In addition, as high quality garments often means higher production costs and therefore higher selling prices [11], high prices were also included in the list (Table 1). We aimed to survey Chinese consumers' attitude towards these factors and examine how much they would be willing to keep a fashion item that possessed one or more of these characteristics. Understanding the reasons why consumers may keep a certain fashion item is paramount for enhancing garment lifetimes and reducing waste, but due to the exploratory nature of the question, we did not form any hypotheses.

**Table 1.** Garment characteristics generated from slow fashion factors.

| Slow Fashion Factors | Garment Characteristics * |
|---|---|
| Quality | High quality fabric<br>Excellent workmanship |
| Design | High aesthetic values<br>Effective and functional design<br>Classic/timeless design<br>High versatility<br>Ease of matching with other clothes<br>Ease of maintenance |
| Sustainability | Use of green or recycled materials |
| Ethicality | Fair trade |
| Local, cultural heritage and craft | Local production<br>Cultural heritage, such as the use of traditional crafts<br>Special production methods, such as handmade |
| Price | High prices |

* Short definitions and examples given in the questionnaire are available online: https://osf.io/u8gnt/?view_only=ba1f70590aa947618a9775272a9b060.

### 4.2. Methodology

Before the conduction of the main study, we conducted a pilot qualitative study with 38 individuals (22 women, 16 men; Mean age = 38.63 years, SD = 11.81) to examine Chinese consumers' understanding of fast and slow fashions. We first explained the two concepts to the participants, and then asked them to come up with fashion types or brands that they believe to fit the two concepts. Results suggest that Chinese consumers did not have problems understanding fast fashion, as participants readily identified brands such as H&M and Zara as representatives of fast fashion. For slow fashion, nine different types of fashion were identified from the pilot study (see Table 2 for the types and brief explanation of main features).

**Table 2.** Slow fashion types identified in the pilot study.

| Fashion Types | Features |
|---|---|
| 1. Famous international luxury brands | Internationally famous fashion brands; high quality; high price. |
| 2. High quality local brands | Local brands; high quality; more accessible prices compared to international luxury brands. |
| 3. Famous international street fashion brands | Casual, comfortable and easy to wear clothes that are practical and long-lasting. |
| 4. Famous Chinese street fashion brands | Local brands of casual, comfortable and easy to wear clothes; more accessible prices compared to famous international street brands. |
| 5. Outdoor and sports brands | Functional, durable and less influenced by changes in fashion trends. |
| 6. Customised/bespoke clothing | Tailor-made clothes that consumers are involved in the design process; higher price compared to mass-produced products. |
| 7. Local independent designer brands | Local brands; unique in design; usually contain traditional handicrafts and/or cultural elements. |
| 8. Vintage/second-hand clothing | Vintage; second-hand; upcycled clothes. |
| 9. Eco clothing | Use of green materials and sustainable techniques. |

Following the pilot study, an online questionnaire study was conducted between November 2021 and February 2022. We aimed to recruit only urban Chinese consumers as they have significantly higher (both actual and disposable) income than their counterparts in the rural areas [57]. The questionnaire was hosted on WJX.CN platform (an online

questionnaire platform that is widely used by companies, academics and individuals in China). Invitations to participate in the study, along with anonymous links to the questionnaire, were sent via WeChat, a popular Chinese instant messaging, social media and mobile payment app. Inclusion criteria were set as: aged 18 years or above, urban Chinese residents who have an interest in fashion and who can read simplified Chinese. In order to obtain as large and diverse a sample as our resources allowed, we used snowballing as our sampling method. Study invitations and anonymous links were sent to the first author's research contacts in China. Recipients were asked to forward the invitations and links to their contacts who fit the study inclusion criteria, and who then forwarded these to their own networks. Contacts were asked to keep track of the number of links they shared and the study stopped after a total of 1000 links had been shared. The study was approved by the Faculty Human Research Ethics Committee of De Montfort University where the first author was completing a doctoral degree.

### 4.3. Materials and Methods

#### 4.3.1. Participants

The 1000 study invitations resulted in 763 valid responses. Of these, about two-thirds (484, or 63.4%) were women; slightly above half (392, or 51.4%) were young adults aged between 18 and 30; and half held an undergraduate university degree (386, or 50.6%). Table 3 reports the detailed demographic information of the sample and Table 4 shows the tabulation of age groups by annual income levels for men and women.

**Table 3.** Demographic information.

| Demographics | | N | % |
| --- | --- | --- | --- |
| Sex | Men | 279 | 36.6% |
| | Women | 484 | 63.4% |
| Age | 18–30 | 392 | 51.4% |
| | 31–50 | 325 | 42.0% |
| | 50+ | 46 | 6.0% |
| Education | High school or below | 222 | 29.1% |
| | Undergraduate degree | 386 | 50.6% |
| | Post-graduate degree | 155 | 20.3% |
| Marital Status | Single | 237 | 31.0% |
| | In long-term relationship | 132 | 17.3% |
| | Married | 374 | 49.0% |
| | Divorced/Widowed | 20 | 2.7% |
| N of Children | None | 405 | 53.1% |
| | One | 204 | 26.7% |
| | Two | 131 | 17.2% |
| | Three | 16 | 2.1% |
| | Four or above | 7 | 0.9% |
| Annual Income (RMB) | Below 80 K | 318 | 41.7% |
| | 80–150 K | 205 | 26.9% |
| | 150–200 K | 71 | 9.3% |
| | 200–400 K | 94 | 12.3% |
| | 400+ K | 75 | 9.8% |
| Occupation | Full-time/Part-time students | 157 | 20.6% |
| | Full-time/Part-time workers | 559 | 73.2% |
| | Full-time homemakers/Retired/Unemployed/Between-jobs | 47 | 6.1% |

**Table 4.** Tabulation of age groups by annual income levels.

| Men (N = 279; 36.6%) | | Annual Income (RMB) | | | | | |
|---|---|---|---|---|---|---|---|
| | | Below 80 K | 80–150 K | 150–200 K | 200–400 K | 400+ K | Total |
| Age | 18–30 | 68 | 39 | 11 | 20 | 13 | 151 |
| | 31–50 | 19 | 25 | 15 | 20 | 24 | 103 |
| | 51+ | 5 | 4 | 2 | 5 | 9 | 25 |
| Total | | 92 | 68 | 28 | 45 | 46 | |
| Women (N = 484; 63.4%) | | Annual Income (RMB) | | | | | |
| | | Below 80 K | 80–150 K | 150–200 K | 200–400 K | 400+ K | Total |
| Age | 18–30 | 153 | 55 | 11 | 15 | 7 | 241 |
| | 31–50 | 58 | 80 | 29 | 34 | 21 | 222 |
| | 51+ | 15 | 2 | 3 | 0 | 1 | 21 |
| Total | | 226 | 137 | 43 | 49 | 29 | |

### 4.3.2. Procedure

The study invitations sent to respondents described the purposes of the study briefly. If a respondent was interested in the study, they could click on the anonymous link in the study invitation. This would take them to the study information sheet which explained the aims and purposes of the study and the participants' right to anonymity, confidentiality and withdrawal. Upon checking the informed consent box, the respondent indicated their agreeing to participating in the study voluntarily, without monetary compensation. The study then started by first presenting questions on well-being (Part 1), demographic questions (Part 2), consumer consumption of fashion types (Part 3) and garment characteristics that may be conducive for consumers' long-term use (Part 4).

### 4.3.3. Measures

*Consumption of fashion types*. Ten different types of fashion (i.e., fast fashion and the nine types of slow fashion listed in Table 2) were presented in the questionnaire. Participants were asked 'What types of fashion do you like to buy?' and were allowed to pick multiple fashion types. All the 'no' answers were dummy-coded as 0 and 'yes' as 1 in all subsequent analyses.

*Well-being.* Butler and Kern's PERMA-Profiler [58] was used to measure well-being. This is a multidimensional measure of flourishing, which is based on Selgiman's PERMA model [26]—positive emotion (P), engagement (E), relationships (R), meaning (M) and accomplishment (A). In the present study, we utilised the 15 items that measure the five main domains (3 items per domain). The items were translated into Chinese by the first author, then back-translated into English by an independent translator. The two versions were compared and all discrepancies discussed and resolved. Responses were recorded on 11-point Likert-style scales from 0 ('never' or 'not at all') to 10 ('always' or 'completely'). A higher score indicates a higher functioning in the specific domain. Table 5 lists all the scale items and their factor loadings.

*Garment characteristics*. Participants were asked if they would be 'willing to buy and keep a fashion item for a long period of time' if the item had certain characteristics. Based on the theorisation of slow fashion factors in Section 2.1, 14 possible garment characteristics were provided (see Table 1). A short explanation was given for each characteristic, followed by examples. Details of the explanations and examples (and the English translation) are available at the Open Science Framework: https://osf.io/u8gnt/?view_only=ba1f70590 aa947618a9775272a9b060. Participants were asked to indicate their willingness due to the characteristics on a five-point Likert scale, from 'very unwilling' (1) to 'very willing' (5).

**Table 5.** Measures of well-being (with scale items and factor loadings).

| PERMA-Profiler [48]—Five Domains of Well-Being (with Scale Items) | Factor Loadings |
| --- | --- |
| **Positive emotion** | |
| In general, how often do you feel joyful? | 0.90 |
| In general, how often do you feel positive? | 0.88 |
| In general, to what extent do you feel content? | 0.88 |
| **Relationship** | |
| To what extent do you receive help and support from others when you need it? | 0.84 |
| To what extent do you feel loved? | 0.80 |
| How satisfied are you with your personal relationships? | 0.86 |
| **Engagement** | |
| How often do you become absorbed in what you are doing? | 0.88 |
| In general, to what extent do you feel excited and interested in things? | 0.83 |
| How often do you lose track of time while doing something you enjoy? | 0.64 |
| **Meaning** | |
| In general, to what extent do you lead a purposeful and meaningful life? | 0.93 |
| In general, to what extent do you feel that what you do in your life is valuable and worthwhile? | 0.91 |
| To what extent do you generally feel you have a sense of direction in your life? | 0.90 |
| **Achievement** | |
| How much of the time do you feel you are making progress towards accomplishing your goals? | 0.89 |
| Howoften do you achieve the important goals you have set for yourself? | 0.88 |
| How often are you able to handle your responsibilities? | 0.81 |

Note: Eleven-point Likert scale from 0 ('never' or 'not at all') to 10 ('always' or 'completely').

## 5. Data Analysis Strategy

Before hypothesis testing, the psychometric properties of the five domains of well-bieng were examined. Internal validity was indicated by Cronbach's Alpha, and following Hair et al. [59], we adopted a value of 0.70 as indication of good validity, and 0.60 as acceptable. Composite reliability was calculated to give another measure of internal consistency in scale items, and following Netemeyer and colleagues' recommendation [60], a threshold of 0.80 was adopted. Convergent validity was indicated by Average Variance extracted (AVE) scores, and the threshold was set at 0.50 [59]. Discriminant validity was assessed by the Fprnell and Larcker Criterion, which states that the square root of AVE for a particular construct should be greater than its correlation with all other constructs.

To test the relationships between slow fashion and the five domains of well-being, the average score of the multiple items was first calculated for each domain and used in correlation analysis and multiple regression analysis. In order to control for the influence of demographic variables that are known to affect well-being [61,62], age groups, gender and annual income groups were dummy coded and entered in Step 1, Step 2 and Step 3, respectively. Fast fashion and slow fashion types were entered in Step 4. Of the three age groups, young adults aged between 18 and 30 years were used as the reference group for comparison with adults aged between 31 to 50, and those who were aged 51 or above. For gender, women (coded as 1) were compared to men (coded as 0). For annual income, as the per capita annual income of Chinese urban residents in the top 30 cities in China in 2021 was RMB186,706.40 [63], we used the annual income group RMB150–200 K as the reference group and compared them to two low income groups—those who earned below RMB80 K and those who earned between RMB80–150 K—and two high earner groups—those who had an annual income between RMB200–400 K, and those whose annual income exceeded RMB400 K.

For garment characteristics that may enhance long-term use, we first described the central tendency of consumers' preferences by calculating the percentage of participants who chose different characteristics. Repeated measures ANOVA with a Greenhouse-Geisser

correction and post hoc analysis with a Bonferroni adjustment were used to test whether consumers' willingness was significantly different across 14 garment characteristics.

## 6. Results

### 6.1. Psychometric Properties of Well-Being Latent Variables

Table 6 reports the psychometric properties of the five latent variables of well-being, and the inter-correlations. Conbrach's Alphas and Composite reliability for the five latent variables all meet the required threshold, suggesting good internal validity and consistency. AVE scores are all above 0.50, suggesting convergent validity, and the square root of AVE scores are all greater than the corresponding correlations, demonstrating satisfactory discriminant validity.

**Table 6.** Psychometric properties of the well-being variables and intercorrelations.

|  | P | E | R | M | A |
|---|---|---|---|---|---|
| Positive emotion | – | 0.66 *** | 0.73 *** | 0.80 *** | 0.73 *** |
| Engagement |  | – | 0.54 *** | 0.68 *** | 0.69 *** |
| Relationship |  |  | – | 0.66 *** | 0.58 *** |
| Meaning |  |  |  | – | 0.85 *** |
| Achievement |  |  |  |  | – |
| Cronbach's Alpha | 0.87 | 0.64 | 0.78 | 0.90 | 0.83 |
| Composite reliability | 0.92 | 0.81 | 0.96 | 0.94 | 0.90 |
| Average Variance Extracted (AVE) | 0.79 | 0.59 | 0.69 | 0.83 | 0.74 |
| $\sqrt{}$ AVE | 0.89 | 0.77 | 0.83 | 0.91 | 0.86 |

Note: Eleven-point Likert scale from 0 ('never' or 'not at all') to 10 ('always'); *** Correlation is significant at the 0.001 level (2-tailed).

### 6.2. Hypotheses Testing

Table 7 reports the correlations between fashion types and the five well-being variables. Consumption of fast fashion brands was negatively correlated with positive emotion, engagement, meaning and achievement. The various slow fashion types' correlations with the well-being dimensions were more varied. In general, slow fashion types, particularly high quality local brands, and customised/bespoke clothing, were positively correlated with various well-being variables. These results provided some support to the hypotheses concerning the general negative relationships well-being has with fast fashion and positive relationships with slow fashion. One notable exception however is vintage/second-hand clothing, which was negatively correlated with positive emotion.

**Table 7.** Correlations between fashion types and well-being domains.

|  | P | E | R | M | A |
|---|---|---|---|---|---|
| Fast fashion brands | −0.10 ** | −0.11 ** | −0.03 | −0.15 ** | −0.17 ** |
| Famous international luxury brands | 0.04 | 0.08 * | 0.11 ** | 0.07 | 0.05 |
| High quality local brands | 0.08 * | 0.02 | 0.10 ** | 0.10 ** | 0.09 * |
| Famous international street fashion brands | −0.04 | 0.04 | 0.07 * | −0.02 | −0.03 |
| Famous Chinese street fashion brands | 0.00 | 0.00 | 0.01 | −0.02 | -0.03 |
| Outdoor and sports brands | 0.04 | 0.08 * | 0.00 | 0.05 | 0.06 |
| Customised/bespoke clothing | 0.04 | 0.10 ** | 0.06 | 0.12 ** | 0.13 ** |
| Local independent designer brands | −0.03 | 0.05 | 0.06 | −0.00 | 0.01 |
| Vintage/second-hand clothing | −0.09 * | −0.03 | −0.03 | −0.04 | −0.03 |
| Eco clothing | 0.07 * | 0.08 * | 0.01 | 0.05 | 0.05 |

Notes: P = positive emotion; E = engagement; R = relationship; M = meaning; A = accomplishment; ** Correlation is significant at the 0.01 level (2-tailed); * Correlation is significant at the 0.05 level (2-tailed).

Table 8 reports all the standardised coefficients, the *t*-values and *p*-values of the predictors on the five well-being domains in multiple regression. For hedonic well-being 'positive emotion', even though the model was significant, $F(17, 745) = 3.72$, $p < 0.001$, none of the fashion types emerged as significant predictors. Hypotheses 1 and 2 were therefore rejected.

**Table 8.** Standardised coefficients and significance of the predictors in the multiple regression analyses.

| DV | IV | $\beta$ | $t$ | $p$ |
|---|---|---|---|---|
| Positive emotion | [Control variables] | | | |
| | Age | | | |
| | 31–50 years | 0.14 | 3.14 | 0.002 |
| | 50+ years | 0.11 | 2.73 | 0.007 |
| | Gender | 0.00 | 0.06 | 0.951 |
| | Income level | | | |
| | below 80 K | −0.14 | −2.12 | 0.034 |
| | 80–150 K | −0.10 | −1.68 | 0.093 |
| | 200–400 K | −0.01 | −0.18 | 0.857 |
| | over 400 K | 0.02 | 0.45 | 0.651 |
| | Fast fashion | −0.04 | −1.18 | 0.237 |
| | Slow fashion | | | |
| | Type 1 | 0.03 | 0.79 | 0.431 |
| | Type 2 | 0.04 | 1.03 | 0.302 |
| | Type 3 | −0.03 | −0.82 | 0.410 |
| | Type 4 | 0.07 | 1.77 | 0.077 |
| | Type 5 | 0.00 | 0.09 | 0.926 |
| | Type 6 | 0.03 | 0.75 | 0.453 |
| | Type 7 | 0.01 | 0.28 | 0.782 |
| | Type 8 | −0.06 | −1.56 | 0.120 |
| | Type 9 | 0.06 | 1.53 | 0.126 |
| Engagement | [Control variables] | | | |
| | Age | | | |
| | 31–50 years | 0.02 | 0.36 | 0.722 |
| | 50+ years | 0.07 | 1.64 | 0.102 |
| | Gender | −0.06 | −1.54 | 0.123 |
| | Income level | | | |
| | below 80 K | −0.19 | −2.91 | 0.004 ** |
| | 80–150 K | −0.10 | −1.66 | 0.098 |
| | 200–400 K | −0.10 | −2.00 | 0.046 * |
| | over 400 K | −0.00 | −0.05 | 0.959 |
| | Fast fashion | −0.07 | −1.79 | 0.074 |
| | Slow fashion | | | |
| | Type 1 | 0.04 | 1.04 | 0.298 |
| | Type 2 | −0.01 | −0.29 | 0.770 |
| | Type 3 | 0.00 | 0.08 | 0.934 |
| | Type 4 | 0.01 | 0.42 | 0.678 |
| | Type 5 | 0.06 | 1.67 | 0.095 |
| | Type 6 | 0.07 | 1.98 | 0.048 * |
| | Type 7 | 0.07 | 1.81 | 0.070 |
| | Type 8 | −0.04 | −1.03 | 0.305 |
| | Type 9 | 0.07 | 1.81 | 0.070 |
| Relationship | [Control variables] | | | |
| | Age | | | |
| | 31–50 years | 0.08 | 1.80 | 0.072 |
| | 50+ years | 0.08 | 1.87 | 0.061 |
| | Gender | 0.05 | 1.23 | 0.221 |
| | Income level | | | |
| | below 80 K | −0.17 | −2.57 | 0.010 ** |
| | 80–150 K | −0.13 | −2.20 | 0.028 * |
| | 200–400 K | −0.05 | −0.97 | 0.331 |
| | over 400 K | −0.05 | −0.98 | 0.327 |

**Table 8.** *Cont.*

| DV | IV | β | t | p |
|---|---|---|---|---|
| | Fast fashion | −0.00 | −0.07 | 0.943 |
| | Slow fashion | | | |
| | Type 1 | 0.07 | 1.82 | 0.069 |
| | Type 2 | 0.05 | 1.23 | 0.220 |
| | Type 3 | 0.04 | 1.02 | 0.307 |
| | Type 4 | 0.02 | 0.47 | 0.640 |
| | Type 5 | −0.01 | −0.27 | 0.791 |
| | Type 6 | 0.03 | 0.93 | 0.352 |
| | Type 7 | 0.05 | 1.30 | 0.192 |
| | Type 8 | −0.03 | −0.82 | 0.411 |
| | Type 9 | 0.00 | 0.06 | 0.952 |
| Meaning | [Control variables] | | | |
| | Age | | | |
| | 31–50 years | 0.11 | 2.69 | 0.007 ** |
| | 50+ years | 0.12 | 3.05 | 0.002 ** |
| | Gender | −0.03 | −0.76 | 0.448 |
| | Income level | | | |
| | below 80 K | −0.18 | −2.80 | 0.005 *** |
| | 80–150 K | −0.10 | −1.62 | 0.107 |
| | 200–400 K | −0.02 | −0.34 | 0.734 |
| | over 400 K | 0.05 | 1.08 | 0.282 |
| | Fast fashion | −0.08 | −2.09 | 0.037 * |
| | Slow fashion | | | |
| | Type 1 | 0.03 | 0.88 | 0.377 |
| | Type 2 | 0.04 | 1.17 | 0.242 |
| | Type 3 | −0.04 | −.93 | 0.352 |
| | Type 4 | 0.04 | 1.09 | 0.278 |
| | Type 5 | 0.02 | 0.45 | 0.651 |
| | Type 6 | 0.09 | 2.55 | 0.011 * |
| | Type 7 | 0.03 | 0.75 | 0.453 |
| | Type 8 | −0.01 | −0.25 | 0.800 |
| | Type 9 | 0.03 | 0.78 | 0.437 |
| Achievement | [Control variables] | | | |
| | Age | | | |
| | 31–50 years | 0.11 | 2.49 | 0.013 * |
| | 50+ years | 0.10 | 2.54 | 0.011 * |
| | Gender | −0.04 | −1.15 | 0.250 |
| | Income level | | | |
| | below 80 K | −0.21 | −3.20 | 0.001 *** |
| | 80–150 K | −0.10 | −1.70 | 0.089 |
| | 200–400 K | −0.04 | −0.83 | 0.405 |
| | over 400 K | 0.05 | 1.09 | 0.275 |
| | Fast fashion | −0.10 | −2.83 | 0.005 ** |
| | Slow fashion | | | |
| | Type 1 | −0.01 | 0.34 | 0.734 |
| | Type 2 | 0.03 | 0.94 | 0.348 |
| | Type 3 | −0.04 | −0.96 | 0.338 |
| | Type 4 | 0.03 | 0.87 | 0.383 |
| | Type 5 | 0.02 | 0.60 | 0.552 |
| | Type 6 | 0.10 | 2.21 | 0.007 ** |
| | Type 7 | 0.04 | 0.04 | 0.317 |
| | Type 8 | −0.00 | −0.00 | 0.950 |
| | Type 9 | 0.03 | 0.03 | 0.471 |

Notes: Slow fashion types: 1 = Famous international luxury brands; 2 = High quality local brands; 3 = Famous international street fashion brands; 4 = Famous Chinese street fashion brands; 5 = Outdoor and sports brands; 6 = Customised/bespoke clothing; 7 = Local independent designer brands; 8 = Vintage/second-hand clothing; 9 = Eco clothing. * *t*-test is significant at the 0.05 level (2-tailed); ** *t*-test is significant at the 0.01 level (2-tailed); *** *t*-test is significant at the 0.001 level (2-tailed).

'Engagement' was positively and significantly predicted by one of the slow fashion types, customised/ bespoke clothing, $\beta = -0.19$, $p = 0.004$. The model explained 4.8% of variability in the variable, $F(17, 745) = 3.27$, $p < 0.001$, and demonstrated a small effect, Cohen's $f^2 = 0.05$.

The model for 'meaning' was significant, $F(17, 745) = 5.39$, $p < 0.001$, even though effect size was small, Cohen's $f^2 = 0.10$ and adjusted $R^2 = 0.09$. Fast fashion consumption

was a significant and negative predictor, $\beta = -0.08$, $p = 0.037$, while customised/ bespoke clothing consumption was significant and positive, $\beta = 0.09$, $p = 0.011$.

For 'achievement', the model was significant, $F(17, 745) = 5.81$, $p < 0.001$, with a small effect Cohen's $f^2 = 0.11$ and explained 9.7% of variability. Consumption of fast fashion negatively predicted 'achievement', $\beta = -0.10$, $p = 0.005$, while customised/ bespoke fashion consumption positively predicted it, $\beta = 0.10$, $p = 0.007$.

These results therefore supported Hypotheses 3 and 4, which state that slow fashion consumption positively, and fast fashion consumption negatively, predict the three eudaimonic well-being domains: 'engagement', 'meaning', and 'achievement'.

### 6.3. Garment Characteristics That Enhance Consumers' Willingness for Long-Term Use

Figure 1 reports the percentage of participants' responses concerning garment characteristics. The 14 characteristics could be arranged into four main groups in terms of consumers' willingness to keep the garment for long term use. The first group comprised three characteristics from the design and quality factors of slow fashion: ease of matching, classic/timeless design and ease of maintenance. They were rated most highly by consumers, with over 75% of participants reportedly being very willing or willing to keep a garment with these characteristics for long term use. We compared the means of reported willingness (on a five-point Likert-scale) by a repeated measure ANOVA with a Greenhouse-Geisser correction and found that consumers' willingness did not differ across these three characteristics, $F(2, 1442.79) = 1.25$, $p = 0.286$, $\eta p^2 = 0.002$.

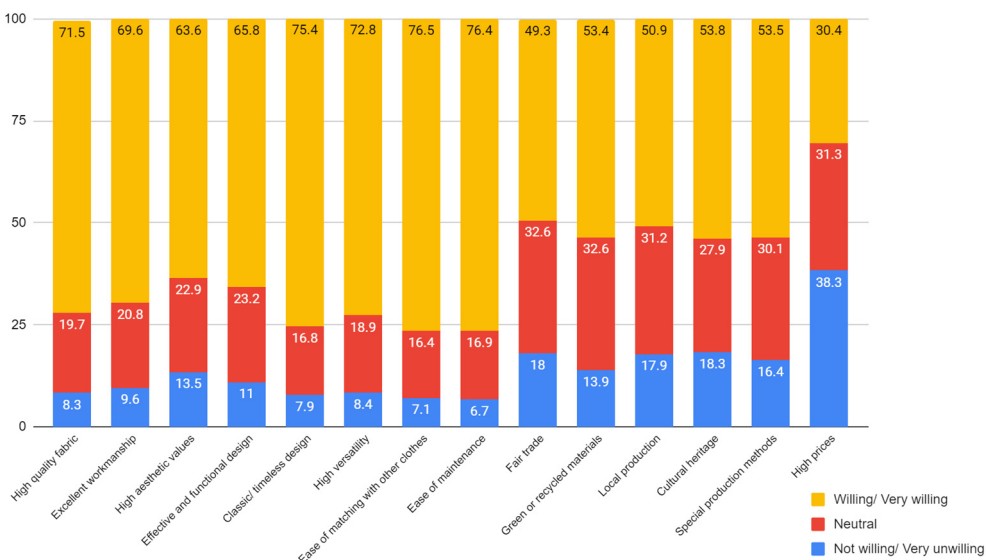

**Figure 1.** Willingness to keep a fashion item for long-term use due to 14 garment characteristics (in %).

The second group was made up of another three design and quality factors of slow fashion: versatility, fabric quality and excellent workmanship. They were the 4th to 6th highest rated characteristics, though willingness levels were significantly lower compared to the top three characteristics, $F(5, 2949.089) = 12.36$, $p < 0.001$, $\eta p^2 = 0.016$. Post hoc analyses with a Bonferroni adjustment further showed that willingness was significantly lower for versatility and fabric quality, compared to that due to ease of matching and classic/timeless design (all $p$s < 0.05). The differences in willingness on versatility, fabric quality and excellent workmanship were not significant, all $p$s > 0.09.

The third group consisted of the remaining design and quality characteristics from slow fashion: effective and functional design and aesthetic values. Consumers' willingness to keep garments due to these two characteristics was significantly lower than those characteristics in the second group, $F(4, 2574.67) = 22.69$, $p < 0.001$, $\eta p^2 = 0.029$, but they induced significantly higher willingness to the fourth group—characteristics related to

ethical, sustainable, and cultural factors of slow fashion, $F(6, 3554.23) = 25.21$, $p < 0.001$, $\eta p^2 = 0.044$.

The price of a garment seemed to be the least important characteristic, with only 30.4% of participants saying they would be 'willing' or 'very willing' to keep the item because of its price. In terms of scores of willingness, it was also significantly lower than the willingness induced by the fourth group of characteristics, $F(5, 2808.27) = 88.30$, $p < 0.001$, $\eta p^2 = 0.104$. Post hoc analyses with a Bonferroni adjustment further showed that willingness due to high price was significantly lower compared to that due to cultural heritage, special production methods, green or recycled materials, local production and fair trade, all $p$s $< 0.001$.

## 7. Discussion

This study aims to systematically evaluate the relationships between different types of fast and slow fashion and individual well-being. In addition, it attempts to explore what garment characteristics can increase consumers' willingness to own and keep clothing items for long-term use. Results showed that consumption of customised/bespoke clothing, a type of slow fashion that emphasises consumers' involvement in the creation process, positively predicted all three eudaimonic well-being domains: 'engagement', 'meaning' and 'achievement'. In contrast, consumption of fast fashion negatively predicted these three domains. Hedonic well-being 'positive emotion' was negatively correlated with fast fashion consumption, and with vintage/second-hand clothing, but these relationships did not reach significant levels when they were evaluated in multiple regression models that controlled for the effects of demographic variables of age, gender and annual income. In terms of garment characteristics, design seemed to be the most important factors in driving willingness for long-term use as consumers selected ease of maintenance, ease of matching, and classic/timeless design as the top three characteristics. This is followed by quality-related considerations such as fabric and workmanship. Next the focus was on aesthetic and cultural elements such as local crafts. Sustainability elements such as use of green or recycled materials, ethical elements such as fair trade were all rated as relatively less important, and price was the least important consideration in consumers' willingness to enhance garment lifetimes.

### 7.1. Fashion Consumption and Well-Being: Theoretical Implications

Previous studies on mass customisation of consumer products have demonstrated many advantages such as satisfying the need for uniqueness and overcoming problems with standard products [64]. However, the positive relationships that customised/bespoke clothing have with the three indicators of eudaimonic well-being, as well as its non-significant relationship with hedonic well-being indicator positive affect, suggest the underlying psychological mechanism is more complex than mere satisfaction with consumption. Norton, Mochon and Ariely show that individuals tend to value their own self-made products, whether assembling mass-produced flat-packed furniture, folding intricate Japanese art of origami, or building sets of Legos models [65]. This phenomenon, dubbed the IKEA effect by the researchers, is observable not only among adults but also in children as young as 5 years old and from diverse backgrounds such as British and Indian cultures [66]. Furthermore, the IKEA effect may have arisen from the psychological need for competence [65], and it could signal an extension of the creator's identity to the self-made product [66]. This is in line with Ryan and Deci's SDT, which considers humans to be active and growth-oriented, and competence is considered as one of the three innate human needs that underlie general well-being [47]. Linking all these to our current findings concerning customised/bespoke clothing and the three domains of eudaimonic well-being ('engagement', 'meaning' and 'achievement'), one can see the plausible mediating role of competence in the relationship as this type of slow fashion actively engages consumers' ideas and inputs, and is therefore better placed to satisfy the need for competence.

If the sense of competence is indeed the underlying psychological mechanism that mediates slow fashion with individuals' eudaimonic well-being, then opportunities to

increase consumers' competence by offering them hands-on experiences in the design or even production process of fashion items could be an avenue for promoting and increasing the popularity of slow fashion. This could be combined with other efforts that enhance sustainability, such as fashion upcycling, a practice that currently is considered more as a niche than as a truly viable alternative to fast fashion. Indeed, a recent study that examines Italian upcyclers' motivations by analysing online posts and comments over an eight-year time span shows that competence acquisition, creativity and autonomy are some of the most prominent reasons that drive upcycling behaviour [67]. Shi et al. [68] interviews with 34 upcyclers in China further collaborate the importance of competence as a motive. What is more, the researchers identify spillover effect on other consumer-focused pro-environmental behaviours as well as swithover effect on purchasing of upcycled products.

In addition, ownership and self-concept, two variables that are closely linked to competence, also need to be considered. Castagna et al. [69] experiments on slow fashion and self-signalling show that slow fashion activates self-concepts of nonconformity, pro-environmental and frugal identities. Importantly, they also identify ownership as an important moderator. When participants were being exposed to an ownership condition in which they could customise their slow fashion clothes, participants experienced higher positive emotion and higher frugality signalling. These results may further explain the positive relationship between customised/bespoke clothing and well-being in our current study.

Apart from customised/bespoke clothing, other slow fashion types did not emerge as significant predictors of well-being. What is particularly noteworthy is that high-end fashion, such as international luxury brands and high quality local brands did not significantly predict any of the well-being indicators. This challenges popular consumers' perception that high-end fashion brands, with their association with prestige and status, can result in increased self-esteem and positive emotions [70,71]. Instead, our findings suggest that fashion items that consumers typically spend a lot of money on do not necessarily make them happier, nor do they contribute to consumers' senses of achievement, meaning and engagement. This is particularly the case when we consider the negative relationship that we found between fast fashion and well-being, which corroborates previous findings on compulsive buying [35–38] and those in the materialism paradigm [39–42,45].

From the perspective of slow fashion promotion, the above findings are encouraging results as they suggest any positive relationship that slow fashion may have with well-being is not likely to be due to high price and exclusivity. From a theoretical point of view, however, it is necessary to examine moderators that may explain the lack of relationships. Theories on the motives of pro-environmental behaviour have long proposed the distinction between self-interested (for example, economic) reasons and self-transcendent causes (such as protecting other living beings and the environment) and their subsequent psychological and behavioural outcomes [72,73]. This distinction in motives or causes could also be seen in fashion consumers' focus. For example, Jun and Jin's [74] cluster analysis of slow fashion consumers in the US shows that a significant portion of individuals are drawn to the exclusivity element of slow fashion, which suggests slow fashion could be construed as a type of status consumption, similar to what researchers in the materialism paradigm theorise. Similarly, the recent trend of collaborative consumption of fashion such as fashion renting and swapping has been shown to relate to vastly different goals that range from self-concerned to environmentally and ethically concerned [75,76]. All these suggest individual differences such as values, and their resulting goals and motives, may not only just affect willingness and actual behaviour in sustainable consumption practice, but may also moderate the psychological link between such consumption practice and their well-being outcomes.

### 7.2. Practical and Social Implications for Sustainability

Even though participants from the pilot interview study identified vintage or second-hand clothing as a type of slow fashion, the negative correlation from the main study suggested that Chinese consumers may not consider this type of clothing desirable. Perhaps

this is due to the fact that vintage or second-hand clothing in China is still largely seen as a way to achieve economic savings, and hence as a sign of poverty [77], rather than as means to achieve environmental sustainability or as creative pathways to alternative fashion choices. Despite this general tendency, there are encouraging signs that attitudes are starting to change. In recent years, a number of highly stylish vintage and second-hand clothing shops have opened in big cities such as Beijing and Shanghai, and they tend to be popular among young people [78]. This trend is important because it not only provides much needed retail outlets for independent and local designers and makers, but it increases the visibility and availability of slow fashion products to consumers, as availability has been identified as one of the main challenges consumers faced in choosing to support slow fashion such as vintage or upcycled clothing [79].

The non-significant relationship between eco clothing and well-being, together with the relatively low rating of fair trade, local brands and green materials as garment characteristics that will drive long-term use, suggest that Chinese consumers are not overly concerned with the ethical and ecological aspects of fashion. While at first glance this seems to contradict the extant literature, and the seemingly rising levels of awareness on sustainability in China [80], this is in fact similar to findings from other studies on consumers' motivation on limiting fashion consumption. For example, Wu and colleagues [81] examined 834 individual autobiographies and blog entries from The Great American Apparel Diet, an online forum for individuals who wanted to change their behaviour of overconsumption of fashion. After classifying all the reasons that were given, the researchers concluded that personal motivations such as self-improvement/self-control, change of lifestyle and saving time were the most prominent category of motivations, accounting for 44% of all the reasons given. Environmental or ecological concerns such as reducing eco-footprint and being green only accounted for 10% of overall reasons. Vladimirova's [82] analysis of extensive blog entries from three online minimalist fashion challenges, in which participants limited themselves to wearing a small number of clothing items for a certain period of time (such as three months), yielded very similar results. The researcher particularly noted the absence of explicit environmental or sustainability concerns as part of the motivations or as benefits from joining the challenges, and proposed that people downsize, first and foremost, for self-interested, personal reasons and not to save the planet' (p. 113).

While raising awareness on the importance of sustainability and ethicality is useful, as shown in the emerging influence of conscious fashion consumers and responsible fashion companies [83], our present findings also suggest another pathway to promote slow fashion and to shift consumption behaviours of the general public, and that is, focusing on the practicality and versatility of the design and the materials used. As Fletcher [84] argues, making a product that consumers are willing to use for a long time is different from making a product that is merely physically durable. To enhance garment lifetimes, fashion products need to satisfy consumers' multiple needs at the same time. From a design perspective, clothing that can be worn on different occasions, is easy to match and that can be shared to fit people of different figures can increase its versatility. These styles may be unisex, loose and seamlessly finished making them very versatile. Alternatively, transformable design creates versatile garments from another angle, offering more than two functional and/or alternative aesthetic styles, allowing a garment to be converted into different looks to suit different personal needs and purposes. Therefore, an appropriate design approach could provide a sustainable alternative that extends product lifetime and reduces excessive garment consumption [85]. In addition, classic design, with its physical durability and style longevity that enhances practicality, is often seen as another means of extending garment lifetime [86]. Classic design refers to timeless, enduring and universally appealing style. It often encompasses iconic works of an artist, a style, or a period [87]. In short, if there is a need for a slow fashion approach to sustainability, exploring versatility and better use of design may be a way to achieve this. Finally, the local and cultural heritage aspect such as traditional patterns or crafts, was not one of the top factors that attracted long-term retention, but more than half of the participants were 'willing' or 'very willing'

to choose products with such characteristics. It suggests that this factor still has some potential to attract slow fashion consumers. It is also an important feature of slow fashion that contributes to maintaining cultural diversity, keeping traditional clothing and textile production methods and special crafts alive, giving more possibilities, personality and meaning to the way we dress and make things. Thus it is also worth exploring.

### 7.3. Limitations and Future Direction

The current study is not without limitations. First, as the study relied on convenient sampling, the current sample is not representative of all Chinese urban consumers. The presence of a large number of full-time university students means our sample is overrepresenting the higher education sector, but also lower annual income—as over one fifth of the participants in the current study were university students who reported an annual income of RMB80 K or below, compared to the median income of RMB150 K for urban residents in 30 main Chinese cities [57]. Second, the correlational design of the study does not allow for inference of causality. Given the lack of mediators and moderators, the present study is only able to test general hypotheses concerning how well-being may be positively related with slow fashion. The complex underlying psychological mechanism needs to be unpacked and examined systematically in future studies. Third, the lack of validated measurement of slow fashion as a latent construct also meant we had to utilise fashion types as a somewhat ad hoc measure. Even though the fashion types were generated by a pilot interview study with 38 Chinese consumers, further studies are needed to evaluate the conceptual and discriminatory validity of the measure. As for the relationship between slow fashion and well-being, future studies that preferably employ nationally representative samples are needed to see if the current findings are able to be replicated across different age, education and income groups. Potentially more importantly, experimental studies are needed to investigate the causal and mediating roles of psychological variables of competence and autonomy, in the relationships between slow fashion such as bespoke fashion and eudaimonic well-being such as senses of engagement and achievement.

### 7.4. Contributions and Conclusion

The slow-fashion and well-being link has been implicated in small-scale qualitative studies that are based on predominantly white participants from western societies [10,45]. The present research provides the first quantitative evidence to ascertain the positive relationship between the two among Chinese apparel consumers. Coupled with the findings that suggest Chinese consumers are more willing to keep garments for long-term use if they are of good quality and timeless design, the current research provides both theoretical and practical insights into possible pathways for promoting slow fashion consumption. By incorporating consumer well-being into the design processes of apparel consumption, and by paying attention to enhancing not only quality but also functional designs that can withstand the changes in fashion trends, the fashion industry can actively encourage consumers' behavioural changes in order to achieve sustainability.

**Author Contributions:** Conceptualization, A.L., E.B. and L.K.; methodology, A.L., E.B. and L.K.; formal analysis, L.K. and A.L.; investigation, A.L.; resources, A.L.; data curation, A.L.; writing—original draft preparation, L.K. and A.L.; writing—review and editing, L.K., A.L. and E.B.; supervision, E.B. and L.K.; project administration, A.L.; funding acquisition, A.L. All authors have read and agreed to the published version of the manuscript.

**Funding:** This research was partially supported by the research grant 'Young Innovative Talents in General Universities of Guangdong Province' (2019WQNCX144) from Department of Education of Guangdong Province, China.

**Institutional Review Board Statement:** The study was conducted in accordance with the Declaration of Helsinki, and approved by the ADH Faculty Ethics Committee of De Montfort University, UK (12 July 2019).

**Informed Consent Statement:** Informed consent was obtained from all subjects involved in the study.

**Data Availability Statement:** Data supporting reported results can be found at: https://osf.io/u8 gnt/?view_only=ba1f70590aa947618a9775272a9b0606.

**Conflicts of Interest:** The authors declare no conflict of interest.

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
