# Peer review of "Slow Fashion Is Positively Linked to Consumers’ Well-Being: Evidence from an Online Questionnaire Study in China"

_sustainability, doi:10.3390/su142113990_

Round 1

Reviewer 1 Report

Dear Authors

On a cursory look, the basic idea behind the research seems good. The authors try to evaluate the relationship between slow fashion and individual well-being among  Chinese consumers. But, as I went deeper, there were a lot of issues. I have provided detailed comments and my concerns. 

1. The title does not sound well. Please rephrase to provide the objectives of the study in a line.

2. First, no studies are cited from 2022. The template used is from 2021. Why?

3. The abstract seems fine. A shorter version is appreciated.

4. The introduction should be in 2-4 paragraphs: background, rationale, RQs, and section plan. Please highlight your RQs in specified sentences preceded by rationales and novelty. Others are okay. Add RQs. 

You have done it in section 3. So there is no need to do anything.

5. LR: Please provide theoretical background first and a literature review after it in two separate sections. More theory background is needed with special emphasis on the constructs and items used in the study and then the relationship (hypotheses). LR needs to be improved. It has to be just reorganized with the inclusion of recent citations.

6. Section-3 is appreciated. Well done.

7. Table-1 & 2: Please provide sources (citation), if possible. If you defined it, say it.

8. Refer: Dash, G., & Paul, J. (2021). CB-SEM vs PLS-SEM methods for research in social sciences and technology forecasting. Technological Forecasting and Social Change173, 121092. (It can be cited) Explore this paper for measurement and analysis.

9. Overall, the method section is a good one. Well done.

10. Results: Same as methods. Good job. One suggestion: You and I can understand technically. But, for a layman and wider audience, always try to write for the audience.

11. The discussion seems okay. Separate Implications into another section. 

12.     Implications must have three components: theory, practice, and social. It can be a separate section.

13.     Future directions can be a separate section along with a conclusion. In short, divide the discussion into three sections as commented above.

14. As the journal is sustainability, your discussion section must have a para linking your findings with sustainability.

15. Thanks for providing the data.

All the best.

Author Response

On a cursory look, the basic idea behind the research seems good. The authors try to evaluate the relationship between slow fashion and individual well-being among Chinese consumers. 

Authors’ response: 

Thank you for the positive comment. 

But, as I went deeper, there were a lot of issues. I have provided detailed comments and my concerns. 

  1. The title does not sound well. Please rephrase to provide the objectives of the study in a line.

Authors’ response: 

Thank you for the comment. We have changed it to ‘Slow fashion is positively linked to consumers’ well-being: Evidence from online questionnaire study in China.’

  1. First, no studies are cited from 2022. The template used is from 2021. Why?

Authors’ response: 

Thank you for the suggestion that we should use the most up-to-date references. As we discussed in the introduction, currently very few studies are focused on the consumption side of slow fashion (p.2). While we could add references about general production aspects of slow fashion, these would do little to enhance the quality of the manuscript. We have, however, cited Domingos and Vale (2022) literature review on slow fashion in the introductory paragraph (p. 2). In terms of literature review on the relationship between slow fashion and individual well-being, there is currently a dearth of research. We have added in results from Domingos et al.’s (2022) literature review (p.4) to illustrate this point. To further address the recommendation of adding references from 2022, we have added in Kim and Oh’ s (2022) study on the relationship between materialism and non-sustainable luxury athleisure products consumption (p.4-5), and Bardey et al’s (2022) study on ‘capsule wardrobe’ (p. 5). In the discussion, we have added in Shit et al.’s (2022) study on upcycling motives (p. 18-19), and Castagna et al.’s (2022) experiments on slow fashion and self-signalling (p. 19). 

As for the template, it was recommended by the journal and was downloaded directly from the journal website. We have to admit we are not knowledgeable whether this was from 2021 or 2022. 

  1. The abstract seems fine. A shorter version is appreciated.

Authors’ response: 

Thank you for the positive comment on the content of the abstract. The word limit for abstract is 200 words, which we adhered to. While a shorter version might be beneficial, it would nevertheless compromise the amount of information that we can summarise. After careful consideration, we decided to retain our original abstract. 

  1. The introduction should be in 2-4 paragraphs: background, rationale, RQs, and section plan. Please highlight your RQs in specified sentences preceded by rationales and novelty. Others are okay. Add RQs. 

You have done it in section 3. So there is no need to do anything.

Authors’ response: 

Thank you for assuring us that no change is needed concerning the introduction and the research questions. We have nevertheless added in a paragraph in the introduction so that the research question is laid out earlier (p.2). 

  1. LR: Please provide theoretical background first and a literature review after it in two separate sections. More theory background is needed with special emphasis on the constructs and items used in the study and then the relationship (hypotheses). LR needs to be improved. It has to be just reorganized with the inclusion of recent citations.

Authors’ response: 

Thank you for the suggestion. In our original manuscript, we first laid out the theoretical background (on slow fashion, and on well-being), and then reviewed the literature on the relationship between the two. Hence, the organisation of the content was exactly the same as the reviewer’s recommendation. In order to make this clearer, we have now reworked the titles of the sections, so they clearly show the theoretical frameworks (Section 2; pp.2-4), followed by the literature review (Section 3; pp. 4-5). 

  1. Section-3 is appreciated. Well done.

Authors’ response: 

Thank you for the positive comment. 

  1. Table-1 & 2: Please provide sources (citation), if possible. If you defined it, say it.

Authors’ response: 

Thank you for the comment. For Table 1, the factors were generated from the definition of slow fashion, as listed in Section 2.1. The description of the table was given in p.7, copied here for ease of reference: 

‘Based on the five main factors of slow fashion that we outlined in Section 2.1, 13 possible characteristics that may contribute to garment lifetimes were generated. In addition, as high quality garments often means higher production costs and therefore higher selling prices [11], high prices were also included in the list (Table 1). We aimed to survey Chinese consumers’ attitude towards these factors and examine how much they would be willing to keep a fashion item that possessed one or more of these characteristics. Understanding the reasons why consumers may keep a certain fashion item is paramount for enhancing garment lifetimes and reducing waste, but due to the exploratory nature of the question, we did not form any hypotheses.’

For Table 2, the slow fashion types were identified by participants in a pilot study prior to the main study. This was explained in Section 4.2 Methodology (p. 7-8). We copied the original text here for ease of reference: 

‘Before the conduction of the main study, we conducted a pilot qualitative study with 38 individuals (22 women, 16 men; Mean age = 38.63 years, SD = 11.81) to examine Chinese consumers’ understanding of fast and slow fashions. We first explained the two concepts to the participants, and then asked them to come up with fashion types or brands that they believe to fit the two concepts. Results suggest that Chinese consumers did not have problems understanding fast fashion, as participants readily identified brands such as H&M and Zara as representatives of fast fashion. For slow fashion, nine different types of fashion were identified from the pilot study (see Table 2 for the types and brief explanation of main features).’ 

  1. Refer: Dash, G., & Paul, J. (2021). CB-SEM vs PLS-SEM methods for research in social sciences and technology forecasting. Technological Forecasting and Social Change, 173, 121092. (It can be cited) Explore this paper for measurement and analysis.

Authors’ response: 

Thank you for the reference. The paper by Dash and Paul (2021) compares covariance-based SEM with partial-least-square-based SEM. This is a very useful paper that gives researchers grounded consideration on which method to use when applying SEM. However, given the confirmatory nature of SEM in testing theoretical models, it is arguably not the best method to employ when the study is more exploratory in nature, as in the case of our present study. Furthermore, given the lack of precise definition and measurement of the concept of slow fashion in the literature (as we mentioned in Section 2.1, pp. 2-3), we are not in a position to analyse the structural relationship between measured variables (i.e., the fashion types) and the latent construct of slow fashion. We have added in a few sentences about this issue in the section on limitations (p.21). 

  1. Overall, the method section is a good one. Well done.

Authors’ response: 

Thank you for the positive comment. 

  1. Results: Same as methods. Good job. One suggestion: You and I can understand technically. But, for a layman and wider audience, always try to write for the audience.

Authors’ response: 

Again, thank you for the positive comment, and the advice on clarity for a wider audience. Due to the comments from the editor, we have updated our results extensively. We have put a large amount of statistical information in tables, while using the text to write short summaries of the results. We hope this arrangement can enhance laypersons’ comprehension.  

  1. The discussion seems okay. Separate Implications into another section. 

Authors’ response: 

Thank you for the positive comment. We have separated the implications into theoretical (pp. 18-19) and practical and social (pp. 19-21).  

  1.     Implications must have three components: theory, practice, and social. It can be a separate section.

Authors’ response: 

Thank you for the suggestion. We believe our discussion of the implications contain these components. 

  1.     Future directions can be a separate section along with a conclusion. In short, divide the discussion into three sections as commented above.

Authors’ response: 

This is now in Section 7.4 Limitations and future direction (p. 21). 

  1. As the journal is sustainability, your discussion section must have a para linking your findings with sustainability.

Authors’ response: 

Thank you for the suggestion. We believe our discussion of the findings is grounded in the context of promoting sustainability, but we have marked the link explicitly by adding in the subsection titled ‘Practical and social implications for sustainability’ (p.19-20). 

  1. Thanks for roviding the data.

Authors’ response: 

We are happy to comply with the best practice of the open science framework. 

Reviewer 2 Report

Thank you for your paper. Some specific observations.

2.1 -Definition of slow fashion seems inaccurate and has assumptions. Please revisit slow fashion

Line 207-209 represent the dimensions of slow fashion better that section 2.1- and researchers may want to considering aligning the terminology used.

Researchers -stay consistent with the term types and dimensions of slow fashion as it is confusing.

 line 75 & 90- check spelling of Poogulangara

line 146-147- daily practice f styling can also bring dissatisfaction

Line 248-252. Is this study divided in two parts? It is like two research questions in one study and they are somewhat different. I recommend separating the two studies, and focus on the well-being for this paper.

Table 1: Source of these factors? Need more research and depth of understanding

Table 2: Who identified the features? Participants or the researchers?

3.3.3 Measures- how the different types of fashion list was developed?

Well-being (line 318) is profiled using PERMA, however, the hypothesis list two types of well-being. Researchers should explain the analysis process here.

Garment characteristics (line 337): How were these identified and listed?

Figure 1: What is the meaning of special production? It is missing the fit which is important for a consumer. I am curious if consumers/participants were given a definition of each characteristic? As some of the characteristics can be interpreted in different ways.

Author Response

Thank you for your paper. Some specific observations.

Authors’ response: 

Thank you for taking the time to review our manuscript and provide insightful comments. 

2.1 -Definition of slow fashion seems inaccurate and has assumptions. Please revisit slow fashion

Line 207-209 represent the dimensions of slow fashion better that section 2.1- and researchers may want to considering aligning the terminology used.

Authors’ response: 

Thank you for the comment. It seems the line numbers may have moved from version to version, but we believe the reviewer may have referred to the following description of slow-fashion that we originally provided on p.6: ‘While it is often considered that slow fashion focuses on quality [13-15], long-term sustainability [16], ethicality [12, 14, 19, 20] and local craft heritage [11, 12, 19, 20]...’

We believe the components listed here -- quality, sustainability, ethicality and local craft heritage -- are the same as what we used in the definition (outlined in Section 2.1). But to make the concept clearer, the definition of slow fashion is now arranged by five main dimensions -- quality, design, sustainability, ethicality and local craft heritage (pp. 2-3). We have also updated the sentence on p.6 to: ‘While it is often considered that slow fashion focuses on quality [10, 13-15], design [16, 17], sustainability [11-20], ethicality [16, 17, 21] and local craft heritage [12, 16, 17, 22]...’  in order to include the design dimension (p.5-6). 

Researchers -stay consistent with the term types and dimensions of slow fashion as it is confusing.

Authors’ response: 

Sorry for the inconsistency. We have gone through the manuscript and standardised the names of slow fashion types. 

 line 75 & 90- check spelling of Poogulangara

Authors’ response: 

Sorry for the mistakes. Due to rewriting of the parts these references are now only in the reference list, and we have double checked to ensure the spelling is correct. Thank you for pointing this out to us. 

line 146-147- daily practice f styling can also bring dissatisfaction

Authors’ response: 

Thank you for the comment. When we write ‘the positive experience of fashion is manifested through the daily practice of styling,’ in no way we are asserting that daily practice of styling only brings satisfaction. Indeed, the effect of daily styling on satisfaction would depend on individuals as well as the social and cultural circumstances. What we are merely attempting to propose is that when fashion brings ‘psychological comfort, satisfaction and comfort’ (p.4), it is often through the action of styling. This notion is based on various studies and scholaristic writings such as Hefferon and Boniwell (2011), Hall (2003) and Lotze (2003). 

Line 248-252. Is this study divided in two parts? It is like two research questions in one study and they are somewhat different. I recommend separating the two studies, and focus on the well-being for this paper.

Authors’ response: 

Thank you for the comment and the suggestion. We believe it is important to retain both parts because they represent two pathways that need to be considered when promoting slow fashion, and hence have important implications. We acknowledge that we should make the link between the two parts clearer in our research question, and should introduce these parts earlier to avoid confusion. We have now added in a new paragraph in the introductory section to highlight this, and introduce our research question in such a way that the link between the two parts is explicit (p.2). 

Table 1: Source of these factors? Need more research and depth of understanding

Authors’ response: 

Thank you for the question. Table 1 lists the characteristics of slow fashion that may enhance garment lifetimes. These characteristics are grounded in the definition of slow fashion that we discuss in Section 2.1. In order to enhance clarity of the links, we have restructured Section 2.1 to include a more detailed explanation of design, sustainability, and ethical elements of slow fashion (p. 2-3). 

Table 2: Who identified the features? Participants or the researchers?

Authors’ response: 

We presume the reviewer was referring to the fashion types listed in Table 2, which were generated from the pilot study. Thirty-eight participants were interviewed by the first author on their fashion consumption habits. They were given a brief explanation of slow fashion, and were asked to think of examples or brands that represent slow fashion. Based on the answers the participants gave, the researchers summarised them into 9 types of slow fashion. This procedure is described in the section on methodology (p. 7-8). 

3.3.3 Measures- how the different types of fashion list was developed?

Authors’ response:

The procedure is described above, as well as in the methodology section (p.7-8). 

Well-being (line 318) is profiled using PERMA, however, the hypothesis list two types of well-being. Researchers should explain the analysis process here.

Authors’ response:

Thank you for the comment. As we outlined in Section 2.2 (p. 3-4), Selgiman’s PERMA model encompases three main components -- the hedonic (i.e., Positive Emotion), the eudaimonic (i.e., Engagement, Meaning, and Achievement), and the relational (i.e., Relationship). Based on past literature that we review in Section 3 (pp. 4-5), it is possible that fast and slow fashion consumption may be related to hedonic and eudaimonic components of well-being, hence our hypotheses. In our hypotheses, we do not only just hypothesise two types of well-being; instead we hypothesise specific well-being domains (i.e., Positive emotion, Engagement, Meaning, and Achievement). We have also added in a short explanation why we do not hypothesise about Relationship (p.6-7). 

Garment characteristics (line 337): How were these identified and listed?

Authors’ response: 

These characteristics were generated from the definition of slow fashion that we discuss in Section 2.1. They are also listed in Table 1, and the rationale explained in Section 4.1 (p. 6-7). 

Figure 1: What is the meaning of special production? It is missing the fit which is important for a consumer. I am curious if consumers/participants were given a definition of each characteristic? As some of the characteristics can be interpreted in different ways.

Authors’ response:

The 14 garment characteristics in Figure 1 are also listed in Table 1. Due to space limitation in Figure 1, we shorten the characteristics to ‘special production methods.’ In Table 1 we explain that this characteristic is under the slow fashion factor ‘local cultural heritage and craft,’ and it refers to methods such as handmade (as opposed to mass/machine produced). 

The garment characteristics were based on slow fashion factors that we discussed in the theoretical framework Section 2.1. Even though fit is an important factor, it was not exclusively limited to slow fashion (presumably fit is important for fast fashion as well) and therefore was not included. 

In the questionnaire, all garment characteristics were given a brief definition. For example, ‘Fair trade’ was defined as ‘the product is from a fair trade company, which meets fair standards of labour and environmental protection.’ The original questionnaire (in Chinese) and its English translation is available on the Open Science Framework: https://osf.io/u8gnt/?view_only=ba1f70590aa947618a9775272a9b060. This link and information is also given in the manuscript (p. 10). 

Round 2

Reviewer 2 Report

Thank you for making the suggested changes/responding to clarify